# Characteristics and outcomes of patients with symptomatic chronic myocardial injury in a Tanzanian emergency department: A prospective observational study

**Faraan O. Rahim**[1]*, **Francis M. Sakita**[2], **Lauren A. Coaxum**[3], **Godfrey L. Kweka**[2], **Zak Loring**[4], **Jerome J. Mlangi**[2], **Sophie W. Galson**[1,3], **Tumsifu G. Tarimo**[2], **Gloria Temu**[5], **Gerald S. Bloomfield**[1,4], **Julian T Hertz**[1,3]

1 Duke Global Health Institute, Duke University, Durham, North Carolina, United States of America,
2 Department of Emergency Medicine, Kilimanjaro Christian Medical Centre, Moshi, Tanzania, 3 Department of Emergency Medicine, Duke University School of Medicine, Durham, North Carolina, United States of America, 4 Department of Internal Medicine, Duke University School of Medicine, Durham, North Carolina, United States of America, 5 Department of Internal Medicine, Kilimanjaro Christian Medical Centre, Moshi, Tanzania

* faraan.rahim@duke.edu

**Data Availability Statement:** All relevant data are within the paper and its Supporting Information files.

## Abstract

### Background

Chronic myocardial injury is a condition defined by stably elevated cardiac biomarkers without acute myocardial ischemia. Although studies from high-income countries have reported that chronic myocardial injury predicts adverse prognosis, there are no published data about the condition in sub-Saharan Africa.

### Methods

Between November 2020 and January 2023, adult patients with chest pain or shortness of breath were recruited from an emergency department in Moshi, Tanzania. Medical history and point-of-care troponin T (cTnT) assays were obtained from participants; those whose initial and three-hour repeat cTnT values were abnormally elevated but within 11% of each other were defined as having chronic myocardial injury. Mortality was assessed thirty days following enrollment.

### Results

Of 568 enrolled participants, 81 (14.3%) had chronic myocardial injury, 73 (12.9%) had acute myocardial injury, and 412 (72.5%) had undetectable cTnT values. Of participants with chronic myocardial injury, the mean (± sd) age was 61.5 (± 17.2) years, and the most common comorbidities were CKD (n = 65, 80%) and hypertension (n = 60, 74%). After adjusting for CKD, thirty-day mortality rates (38% vs. 36%, aOR 1.03, 95% CI: 0.52–2.03, p = 0.931) were similar between participants with chronic myocardial injury and those with acute myocardial injury, but significantly greater (38% vs. 13.6%, aOR 3.63, 95% CI: 1.98–

**Funding:** This study was funded by grants from the Society for Academic Emergency Medicine, the US National Institutes of Health National Heart Lung and Blood Institute [K23-HL155500], Roche Diagnostics, and the Duke Global Health Institute all awarded to JTH. The funders had no role in the study design, study conduct, data analysis, or decision to publish.

**Competing interests:** The authors have read the journal's policy and have the following competing interests: Julian Hertz's institution, Duke University School of Medicine, received a research grant from Roche Diagnostics. Zak Loring's institution, Duke University School of Medicine, received grant support from Huxley Medical and Boston Scientific for unrelated research projects in which he serves as an investigator. There are no patents, products in development or marketed products associated with this research to declare. This does not alter our adherence to PLOS ONE policies on sharing data and materials.

6.65, p<0.001) among participants with chronic myocardial injury than those with undetectable cTnT values.

## Conclusion

In Tanzania, chronic myocardial injury is a poor prognostic indicator associated with high risk of short-term mortality. Clinicians practicing in this region should triage patients with stably elevated cTn levels in light of their increased risk.

## Introduction

Chronic myocardial injury is a condition where patients present with persistently elevated cardiac troponin (cTn) values without overt acute myocardial ischemia [1, 2]. It is now recognized in the Fourth Universal Definition of Myocardial Infarction as a separate entity from the five Myocardial Infarction (MI) subtypes. In chronic myocardial injury, cTn levels are persistently but stably elevated, in contrast to acute myocardial injury, where they rise or fall [3]. Representing up to 58% of cases of elevated cTn values [4], the condition occurs in a heterogenous patient population, having both cardiovascular causes, such as heart failure and cardiomyopathy, and noncardiovascular causes, such as chronic kidney disease (CKD) and toxins [2, 5].

In high-income countries, chronic myocardial injury is associated with high rates of mortality and major adverse cardiac events (MACE), including MI, stroke, and sudden cardiac death [2, 6]. Patients with chronic myocardial injury may even suffer mortality rates that match or exceed those of patients with Type 1 MI, with excess mortality often attributed to noncardiovascular causes [4, 7]. Hence, while chronic myocardial injury represents a heterogenous group of conditions and is pathophysiologically distinct from acute Type I MI, there is increasing recognition that the presence of persistently elevated cTn levels is a poor prognostic indicator.

As the burden of disease in sub-Saharan Africa (SSA) transitions away from communicable diseases and towards non-communicable ones [8], cardiovascular disease (CVD) is becoming a leading cause of morbidty and mortality in the region [9]. Hypertension, heart failure, and MI are associated with high rates of MACE and mortality in SSA [10–13], but little is known about chronic myocardial injury in the region. Although a cross-sectional study from South Africa examined the incidence of acute myocardial injury among patients after non-cardiac surgery [14], to our knowledge, there are no published data on the prognoses of patients with chronic myocardial injury in SSA. To address this gap in the literature, we prospectively described patients with chronic myocardial injury presenting to an emergency department in northern Tanzania and compared their thirty-day mortality rates to those of patients without chronic myocardial injury.

## Methods

### Study setting

This study was conducted at the Kilimanjaro Christian Medical Centre (KCMC). Located in the urban center of Moshi, Tanzania, KCMC is a regional tertiary care hospital that serves over 15 million people across the northern regions of the country. The KCMC emergency department (ED) operates 24 hours a day and is staffed by a mix physicians, clinical officers, and

nurses. It receives all patients who present for unscheduled medical care with or without a referral. The KCMC ED is equipped with laboratory-based and point-of-care cTn assays but does not have the capacity for cardiac catheterization.

## Participant selection

Enrollment for this study was conducted from 8 AM to 11 PM, 7 days per week, from November 5th, 2020 through January 31st, 2023. All adult patients (age ≥ 18 years) who presented to the KCMC ED with chest pain or shortness of breath as a primary or secondary complaint during this enrollment period were eligible for study participation. Exclusion criteria were chest pain secondary to trauma and self-reported fever. Trained research assistants offered enrollment to eligible patients and obtained written, informed consent from those interested in participating.

## Study procedures

Following enrollment, trained research assistants administered a modified version of the World Health Organization (WHO) STEPS survey for non-communicable disease risk factor surveillance [15]. This standardized survey gathered information about sociodemographic background, medical history, and lifestyle behavior from participants. Research assistants also measured and recorded participant height, weight, pulse, blood pressure, and point-of-care glucose. For participants who reported no known history of HIV, a point-of-care HIV test was performed using the Standard Diagnostics Bioline HIV 1/2 assay (Standard Diagnostics, Suwon, Korea). Counseling accompanied all HIV testing, and any patients testing positive for HIV were linked to appropriate follow-up care. A resting twelve-lead electrocardiogram (ECG) was obtained during initial enrollment using the tablet-based PADECG (Edan Instruments, Shenzhen, China). Point-of-care troponin T (cTnT) assays were obtained from all participants at time of enrollment using the Roche cobas h 232 point-of-care system (Roche Diagnostics, Basel, Switzerland). If the initial cTnT value was abnormally elevated (as defined below), a repeat cTnT point-of-care assay was performed three hours later. If obtained by the clinical team, serum creatinine levels were also recorded directly from participants' electronic medical records. Research assistants also collected primary and secondary ED diagnoses directly from the medical record. If the clinical team obtained an echocardiogram on any enrolled participant during their hospital stay, the echocardiogram results were collected from the electronic medical record by the research team.

At time of hospital discharge, research assistants collected final primary and secondary discharge diagnoses from the electronic medical record. For participants discharged home directly from the ED, the final primary and secondary discharge diagnoses documented by the ED physician were used as the hospital discharge diagnoses. In case of in-hospital death, clinician-determined cause of death was collected from the medical record. Thirty days following initial enrollment, participants were contacted via telephone to assess vital status; home visits were conducted for participants unreachable by phone. In case of participant death after hospital discharge, a brief verbal autopsy was performed with a relative, guided by the WHO verbal autopsy instrument [16]. Verbal autopsy data were reviewed by a committee of physicians from Tanzania and the United States to adjudicate causes of death.

## Study definitions

An abnormally elevated cTn was defined as a cTnT value above >40 ng/L (the lower limit of the assay's measuring range, which is above the 99th percentile upper reference limit). Given the manufacturer-defined coefficient of variation of 11%, participants with an abnormally

elevated cTnT value and a subsequent three-hour cTnT value within 11% of the initial value were defined as having chronic myocardial injury. Participants with three-hour delta cTnT values ≥11% of their initial value were defined as having acute myocardial injury. Patients with undetectable troponin values (≤40 ng/L), were defined as being without myocardial injury.

Resting tachycardia was defined as heart rate ≥100 beats per minute. Elevated blood pressure was defined as measured systolic blood pressure ≥ 140 mmHg or diastolic blood pressure ≥ 90 mmHg at enrollment. Hyperglycemia was defined as fasting glucose >126 mg/dl or random glucose >200 mg/dl, as per the American Diabetes Association guidelines [17]. Diabetes was defined by self-reported history of diabetes or measured hyperglycemia. Participant self-report was used to determine history of heart failure, hypertension, MI, stroke, hyperlipidemia, alcohol use, and tobacco use. CKD was defined by participant self-report of CKD or estimated glomerular filtration rate (GFR) < 60/ml/min/1.73 m$^2$ [18]. Estimated GFR was calculated from measured serum creatinine level using the 2021 CKD-EPI Creatinine Equation [19]. Obesity was defined as measured body mass index (BMI) ≥30 kg/m$^2$, per U.S. Centers for Disease Control guidelines [20], and sedentary lifestyle as <150 minutes of self-reported moderately vigorous exercise per week, per WHO guidelines [21]. In addition, self-reported history of MI or stroke in a first-degree relative was used to define family history of MI or stroke. HIV infection was defined by participant self-report or a positive point-of-care HIV test result.

## Chronic myocardial injury etiology adjudication

Physician adjudicators (LAC, ZL, SWG, GSB, JTH), trained in emergency medicine or cardiology, determined etiologies of chronic myocardial injury among patients. Adjudicators independently reviewed medical data that included patient demographics, vital signs, symptoms, comorbidities, results of diagnostic testing (including laboratory tests and echocardiography findings, when available), in-hospital treatments, and physician-documented diagnoses, but they were blinded to individual patient outcomes. Adjudicators were then asked to select an etiology for each patient's chronic myocardial injury from the following list, adapted from the Fourth Universal Definition of Myocardial Infarction guidelines and the American College of Cardiology Scientific Expert Panel recommendations [3, 5]: CKD, heart failure, other cardiomyopathies, hypertension, persistent arrythmias, pulmonary hypertension, toxins (including cardiotoxic chemotherapy), myocarditis, skeletal myopathies, sepsis, pulmonary embolism, or other. When adjudicators did not feel there was enough clinical data to support a specific cause of chronic myocardial injury, the etiology was recorded as "indeterminant." In cases of disagreement between the first two independent adjudicators, a third physician adjudicator served as the tiebreaker.

## Statistical analyses

Our primary objective was to describe characteristics of chronic myocardial injury patients and assess thirty-day mortality rates among patients with and without chronic myocardial injury. Participants who had an initially elevated cTnT value but did not have a second cTnT result available due to death in the ED were excluded from the present analysis. Statistical analyses were performed in Microsoft Excel and the R suite. BMI was calculated directly from measured height and weight. Tanzanian shillings (TSH) were converted to United States dollars (USD) using the World Bank 2021 conversion rate of 1 USD = 2297.76 TSH [22]. Categorical variables are presented as proportions and continuous variables as means with associated standard deviations. Characteristics and thirty-day mortality rates of participants with chronic

myocardial injury were compared with those of participants with acute myocardial injury and those of participants without myocardial injury via Pearson's chi-squared (for categorical variables) and Student's t-test (for continuous variables). Fisher's Exact Test was used when expected cell count was <10. Univariate odds ratios were calculated directly from two-by-two contingency tables. Because of the strong association between chronic myocardial injury and CKD, generalized linear modeling with binomial distribution was utilized to calculate adjusted odds ratios along with their corresponding 95% confidence intervals. This approach addressed the influence of CKD when comparing mortality rates among participants with and without chronic myocardial injury.

## Ethics and data availablity

The institutional review boards at the Tanzania National Institute for Medical Research, KCMC, and Duke Health all reviewed and granted ethical approval for this research study throughout its duration. Additionally, written, informed consent was obtained from all participants prior to enrollment. All data are available in the Supporting Information files.

## Results

During enrollment, 7943 adult patients presenting to the ED were screened for eligibility, and 568 (7.2%) with chest pain or shortness of breath were enrolled (**Fig 1**). Of participants, 81 (14.3%) met the criteria for chronic myocardial injury. Of the 485 (85.4%) without chronic myocardial injury, 412 (84.9%) had undetectable cTnT values, and 73 (15.1%) had acute myocardial injury. Two participants (0.4%) died before a repeat cTnT value was obtained and were therefore excluded from analysis.

**Table 1** presents baseline demographics, vital signs, and pertinent medical history of chronic myocardial injury participants. Of participants with chronic myocardial injury, 45 (56%) were male and the mean (± sd) age was 61.5 (± 17.2) years. Mean (± sd) initial and three-hour cTnT values were 193 (± 258) ng/mL and 192 (± 257) ng/mL, respectively. The

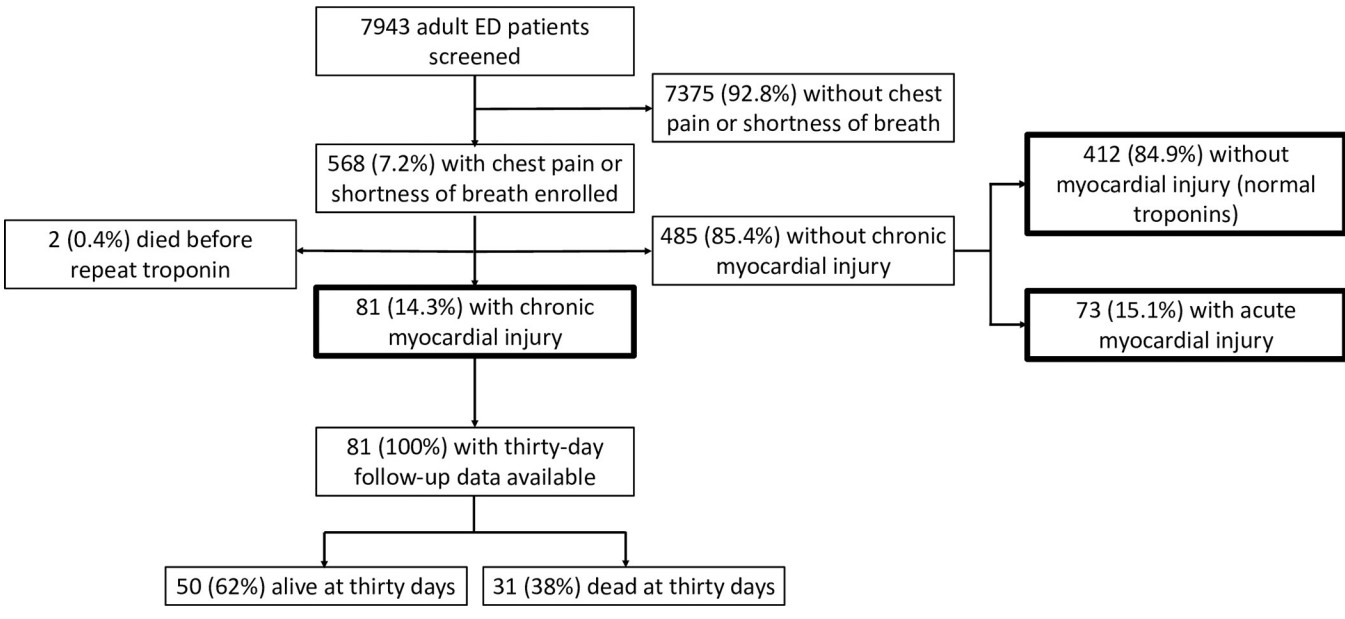

**Fig 1. KCMC patient recruitment and follow-up flowchart.**

**Table 1. Characteristics of chronic myocardial injury patients, northern Tanzania, 2020–2023.**

| Characteristic | Chronic Myocardial Injury Patients (N = 81) | |
|---|---|---|
| | **n** | **%** |
| **Demographics** | | |
| Male Sex | 45 | 56% |
| Age, mean (sd), years | 61.5 (17.2) | |
| Highest level of education attained | | |
| None | 4 | 5% |
| Primary | 54 | 67% |
| Secondary | 8 | 10% |
| University | 15 | 19% |
| Income, mean (sd), USD | 42.23 (81.94) | |
| **Vital Signs / General Health Parameters** | | |
| Heart Rate, mean (sd), bpm | 95 (29) | |
| Systolic BP, mean (sd), mmHg | 145 (35) | |
| Diastolic BP, mean (sd), mmHg | 88 (29) | |
| BMI, mean (sd), kg/m$^2$ | 24.8 (6.0) | |
| Resting Tachycardia | 29 | 36% |
| Elevated Blood Pressure | 48 | 59% |
| Initial Troponin, mean (sd), ng/mL | 193 (258) | |
| Repeat 3-Hour Troponin, mean (sd), ng/L | 192 (257) | |
| Glomerular Filtration Rate (GFR) (mL/min) | | |
| >60 | 13 | 16% |
| 30–60 | 23 | 28% |
| 15–30 | 14 | 17% |
| <15 | 27 | 33% |
| Not Available | 4 | 5% |
| **Comorbidities** | | |
| CKD | 65 | 80% |
| Self-reported history of hypertension | 60 | 74% |
| Self-reported history of heart failure | 29 | 36% |
| Diabetes | 27 | 33% |
| Obese | 15 | 59% |
| Self-reported family history of MI or stroke | 10 | 12% |
| HIV Infection | 4 | 5% |
| Self-reported history of hyperlipidemia | 3 | 4% |
| Self-reported history of MI | 2 | 2% |
| Self-reported history of stroke | 2 | 2% |
| **Daily Medications** | | |
| Antihypertensive | 50 | 62% |
| Antidiabetic | 11 | 14% |
| Aspirin | 9 | 11% |
| Statin | 3 | 4% |
| Clopidogrel | 3 | 4% |
| **Lifestyle/Health Behaviors** | | |
| Sedentary Lifestyle | 78 | 96% |
| Current alcohol use | 20 | 25% |
| Current tobacco use | 9 | 11% |
| Daily fruit and vegetable consumption | 4 | 5% |

most common comorbidities among participants with chronic myocardial injury were CKD (n = 65, 80%), self-reported hypertension (n = 60, 74%), self-reported heart failure (n = 29, 36%), and diabetes (n = 27, 33%). The most common daily medications taken by participants included antihypertensives (n = 50, 62%), antidiabetics (n = 11, 14%), and aspirin (n = 9, 11%). Four (5%) participants had HIV, including 2 participants who self-reported a history of HIV and 2 participants who denied a known history of HIV but tested positive.

Baseline characteristics of patients with chronic myocardial injury were compared with those of 412 patients without myocardial injury (Table 2). Patients with chronic myocardial injury were more likely to be male (55.6% vs. 40.8%, OR 1.81, 95% CI:1.12–2.95, p = 0.001), have CKD (80.2% vs. 25.5%, OR 11.73, 95% CI: 6.64–21.86, p<0.001), report a history of hypertension (74.1% vs. 50.2%, OR 2.81, 95% CI: 1.67–4.90, p<0.001), have diabetes (33.3% vs. 19.7%, OR 2.04, 95% CI: 1.20–3.43, p = 0.007), use antihypertensive medication (61.7% vs. 42.2%, OR 2.20, 95% CI:1.35–3.62, p = 0.001), and earn a lower income (p = 0.014).

**Table 2. Characteristics of patients with chronic myocardial injury vs. patients without myocardial injury, northern Tanzania, 2020–2023.**

| | Chronic Myocardial Injury (N = 81), n (%) | No myocardial injury (N = 412), n (%) | Odds ratio (95% CI) | p |
|---|---|---|---|---|
| Male sex | 45 (55.6) | 168 (40.8) | 1.81 (1.12–2.95) | **0.014*** |
| Post-primary education | 23 (28.4) | 156 (37.9) | 0.65 (0.38–1.09) | 0.105 |
| Resting tachycardia | 29 (35.8) | 136 (33.0) | 1.13 (0.68–1.86) | 0.626 |
| Elevated blood pressure | 48 (59.3) | 205 (49.8) | 1.47 (0.91–2.40) | 0.118 |
| CKD | 65 (80.2) | 105 (25.5) | 11.73 (6.64–21.86) | **<0.001*** |
| Self-reported history of hypertension | 60 (74.1) | 207 (50.2) | 2.81 (1.67–4.90) | **<0.001*** |
| Self-reported history of heart failure | 29 (35.8) | 123 (29.9) | 1.31 (0.79–2.16) | 0.289 |
| Diabetes | 27 (33.3) | 81 (19.7) | 2.04 (1.20–3.43) | **0.007*** |
| Obesity | 15 (18.5) | 91 (22.1) | 0.81 (0.42–1.45) | 0.475 |
| Self-reported family history of MI/Stroke | 10 (12.3) | 79 (19.2) | 0.60 (0.28–1.17) | 0.144 |
| HIV Infection | 4 (4.9) | 12 (2.9) | 1.77 (0.470–5.32) | 0.314 |
| Self-reported history of hyperlipidemia | 3 (3.7) | 16 (3.9) | 0.99 (0.22–3.10) | 0.999 |
| Self-reported history of MI | 2 (2.5) | 13 (3.2) | 0.83 (0.12–3.11) | 0.999 |
| Self-reported history of stroke | 2 (2.5) | 8 (1.9) | 1.35 (0.18–5.65) | 0.672 |
| Daily use of antihypertensive | 50 (61.7) | 174 (42.2) | 2.20 (1.35–3.62) | **0.001*** |
| Daily use of antidiabetic | 11 (13.6) | 54 (13.1) | 1.05 (0.50–2.05) | 0.908 |
| Daily use of aspirin | 9 (11.1) | 34 (8.3) | 1.40 (0.61–2.95) | 0.405 |
| Daily use of statin | 3 (3.7) | 10 (2.4) | 1.60 (0.34–5.47) | 0.457 |
| Daily use of clopidogrel | 3 (3.7) | 24 (5.8) | 0.65 (0.15–1.93) | 0.597 |
| Sedentary lifestyle | 78 (96.3) | 409 (99.3) | 0.19 (0.03–1.13) | 0.059 |
| Current alcohol use | 20 (24.7) | 126 (30.6) | 0.75 (0.42–1.27) | 0.288 |
| Current tobacco use | 9 (11.1) | 40 (9.7) | 1.18 (0.51–2.44) | 0.700 |
| Daily fruit and vegetable consumption | 4 (4.9) | 27 (6.6) | 0.76 (0.22–2.04) | 0.584 |
| | Chronic Myocardial Injury (N = 81), mean (sd) | No myocardial injury (N = 412), mean (sd) | | p |
| Age, years | 61.5 (17.2) | 59.0 (19.4) | | 0.231 |
| Income, USD | 42.23 (81.94) | 139.28 (776.70) | | **0.014*** |
| BMI, kg/m$^2$ | 24.8 (6.0) | 26.2 (6.5) | | 0.055 |
| GFR (mL/min) | 35.0 (32.2)[a] | 67.6 (30.9)[b] | | **<0.001*** |
| Initial Troponin (ng/mL) | 193 (258) | 39 (12) | | **<0.001*** |

[a]Data unavailable for 4/81 participants

[b]Data unavailable for 171/412 participants

**Table 3. Characteristics of patients with chronic myocardial injury vs. acute myocardial injury, northern Tanzania, 2020–2023.**

| | Chronic Myocardial Injury (N = 81), n (%) | Acute Myocardial Injury (N = 73), n (%) | Odds ratio (95% CI) | *p* |
|---|---|---|---|---|
| Male sex | 45 (55.6%) | 43 (58.9) | 0.87 (0.46–1.66) | 0.675 |
| Post-primary education | 23 (28.4) | 25 (34.2) | 0.76 (0.38–1.52) | 0.434 |
| Resting tachycardia | 29 (35.8) | 21 (28.8) | 1.38 (0.70–2.75) | 0.352 |
| Elevated blood pressure | 48 (59.3) | 39 (53.4) | 1.27 (0.67–2.41) | 0.466 |
| CKD | 65 (80.2) | 43 (58.9) | 2.80 (1.38–5.89) | **0.004*** |
| Self-reported history of hypertension | 60 (74.1) | 51 (69.9) | 1.23 (0.60–2.51) | 0.561 |
| Self-reported history of heart failure | 29 (35.8) | 28 (38.4) | 0.90 (0.46–1.74) | 0.743 |
| Diabetes | 27 (33.3) | 28 (38.4) | 0.81 (0.41–1.56) | 0.516 |
| Obesity | 15 (18.5) | 12 (16.4) | 1.15 (0.50–2.72) | 0.735 |
| Self-reported family history of MI/ Stroke | 10 (12.3) | 12 (16.4) | 0.72 (0.28–1.80) | 0.469 |
| HIV Infection | 4 (4.9) | 3 (4.1) | 1.20 (0.24–6.69) | 0.999 |
| Self-reported history of hyperlipidemia | 3 (3.7) | 2 (2.7) | 1.33 (0.20–11.74) | 0.999 |
| Self-reported history of MI | 2 (2.5) | 4 (5.5) | 0.45 (0.05–2.55) | 0.423 |
| Self-reported history of stroke | 2 (2.5) | 3 (4.1) | 0.61 (0.07–4.09) | 0.668 |
| Daily use of antihypertensive | 50 (61.7) | 41 (56.2) | 1.26 (0.66–2.41) | 0.483 |
| Daily use of antidiabetic | 11 (13.6) | 18 (24.7) | 0.48 (0.20–1.10) | 0.079 |
| Daily use of aspirin | 9 (11.1) | 7 (9.6) | 1.17 (0.41–3.51) | 0.757 |
| Daily use of statin | 3 (3.7) | 4 (5.5) | 0.67 (0.12–3.33) | 0.708 |
| Daily use of clopidogrel | 3 (3.7) | 9 (12.3) | 0.28 (0.06–1.02) | **0.046*** |
| Sedentary lifestyle | 78 (96.3) | 72 (98.6) | 0.39 (0.01–3.49) | 0.622 |
| Current alcohol use | 20 (25.0) | 15 (20.5) | 1.26 (0.59–2.75) | 0.540 |
| Current tobacco use | 9 (11.1) | 4 (5.5) | 2.10 (0.64–8.35) | 0.209 |
| Daily fruit and vegetable consumption | 4 (4.9) | 4 (5.5) | 0.90 (0.20–4.11) | 0.999 |
| | Chronic Myocardial Injury (N = 81), mean (sd) | Acute Myocardial Injury (N = 73), mean (sd) | | *p* |
| Age, years | 61.5 (17.2) | 65.7 (15.0) | | 0.112 |
| Income, USD | 42.23 (81.94) | 34.58 (68.36) | | 0.529 |
| BMI, kg/m² | 24.8 (6.0) | 25.0 (5.7) | | 0.785 |
| GFR (mL/min) | 35.0 (32.2)[a] | 37.2 (29.8)[b] | | 0.700 |
| Initial Troponin (ng/mL) | 193 (258) | 374 (556) | | **0.012*** |

[a]Data unavailable for 4/81 participants

[b]Data unavailable for 20/73 participants

Baseline characteristics were also compared between the 81 participants with chronic myocardial injury and the 73 with acute myocardial injury (Table 3). CKD (80.2% vs. 58.9%, OR 2.80, 95% CI: 1.38–5.89, p = 0.004) was more common among participants with chronic myocardial injury, while daily use of clopidogrel (3.7% vs. 12.3%, OR 0.28, 95% CI: 0.06–1.02, p = 0.046) was less common. In addition, participants with chronic myocardial injury were more likely to have a lower initial cTnT value (p = 0.012) than those with acute myocardial injury.

Table 4 describes the care that chronic myocardial injury participants received in the KCMC ED as well as their hospital discharge diagnoses. Most participants with chronic myocardial injury (n = 69, 85%) were admitted for inpatient hospital care after their ED visit. The most common medications administered to participants with chronic myocardial injury in the ED included diruetics (n = 26, 32%), aspirin (n = 22, 27%), clopidogrel (n = 17, 21%), statins

**Table 4. Emergency department management and hospital discharge diagnoses of patients with chronic myocardial injury, northern Tanzania, 2020–2023.**

| | Chronic Myocardial Injury Patients (N = 81) | |
|---|---|---|
| **ED Management** | **n** | **%** |
| Hospitalized | 69 | 85% |
| **Medications Administered** | | |
| Diuretic | 26 | 32% |
| Aspirin | 22 | 27% |
| Clopidogrel | 17 | 21% |
| Statin | 12 | 15% |
| Antibacterial | 11 | 14% |
| Calcium Channel Blocker | 10 | 12% |
| Nitrates | 9 | 11% |
| Calcium | 8 | 10% |
| Heparin | 8 | 10% |
| Bronchodilator | 8 | 10% |
| Steroid | 7 | 9% |
| Analgesic | 6 | 7% |
| Proton Pump Inhibitor | 6 | 7% |
| Beta Blocker | 3 | 4% |
| Antidiabetic | 2 | 2% |
| Hydralazine | 2 | 2% |
| Other | 6 | 7% |
| **Echocardiogram Obtained** | 6 | 7% |
| **Primary Hospital Discharge Diagnosis** | | |
| Heart Failure | 34 | 42% |
| CKD | 15 | 19% |
| Hypertension | 13 | 16% |
| Pneumonia | 8 | 10% |
| COPD | 2 | 2% |
| Pulmonary Embolism | 2 | 2% |
| Atrial Fibrillation | 1 | 1% |
| Cholecystitis | 1 | 1% |
| Urinary Tract Infection | 1 | 1% |
| Plueral Effusion | 1 | 1% |
| Stroke | 1 | 1% |
| Acute Kidney Injury | 1 | 1% |
| Deep Vein Thrombosis | 1 | 1% |
| **Secondary Hospital Discharge Diagnoses[a]** | | |
| Chronic Kidney Disease | 20 | 25% |
| Diabetes | 14 | 17% |
| Hypertension | 12 | 15% |
| Heart Failure | 8 | 10% |
| Electrolyte Imbalance | 7 | 9% |
| Peptic Ulcer Disease/Gastritis | 7 | 9% |
| COPD/Asthma | 4 | 5% |
| Pneumonia | 3 | 4% |
| Hyperlipidemia | 2 | 2% |
| Stroke | 2 | 4% |

*(Continued)*

**Table 4.** (Continued)

| | Chronic Myocardial Injury Patients (N = 81) | |
|---|---|---|
| **ED Management** | **n** | **%** |
| Liver Disease | 2 | 4% |
| Other | 10 | 12% |

[a]Multiple secondary diagnoses possible

(n = 12, 15%), antibacterials (n = 11, 14%), and calcium channel blockers (n = 10, 12%). The most common primary hospital discharge diagnoses included heart failure (n = 34, 42%), CKD (n = 15, 19%), hypertension (n = 13, 16%), and pneumonia (n = 8, 10%); the most common secondary diagnoses were CKD (n = 20, 25%), diabetes (n = 14, 17%), hypertension (n = 12, 15%), and pneumonia (n = 8, 10%). Six (7%) participants with chronic myocardial injury had an echocardiogram performed during their hospital stay; echocardiogram results are summarized in **S1 Table**.

Thirty-day follow-up data was obtained for all 81 participants with chronic myocardial injury; 50 (62%) were alive and 31 (38%) were dead at thirty days. Of the 31 dying within thirty days, 26 (84%) died during the index hospitalization at KCMC, 4 (13%) died at home after hospital discharge, and 1 (3%) died at another hospital after discharge from KCMC. Follow-up data was also obtained for all 73 patients with acute myocardial injury; 47 (64%) were alive and 26 (36%) were dead at thirty days. Follow-up data was obtained for 411 (99.8%) patients without myocardial injury; 1 (0.2%) was lost to follow-up. Of the 411 patients without myocardial injury completing follow-up, 355 (86.4%) were alive and 56 (13.6%) were dead at thirty days.

Thirty-day mortality rates (38% vs. 36%, unadjusted OR 1.12, 95% CI: 0.58–2.17, p = 0.733) were similar between participants with chronic myocardial injury and those with acute myocardial injury. However, the thirty-day mortality rate (38% vs. 13.6%, unadjusted OR 3.92, 95% CI: 2.29–6.65, p<0.001) was significantly greater for participants with chronic myocardial injury than those without myocardial injury.

After adjusting for CKD, thirty-day mortality rates remained significantly greater for participants with chronic myocardial injury than those without myocardial injury (aOR 3.63, 95% CI: 1.98–6.65, p<0.001). Similarly, after adjusting for CKD, there was still no significant difference in thirty-day mortality rates among participants with chronic myocardial injury and those with acute myocardial injury (aOR 1.03, 95% CI: 0.52–2.03, p = 0.931).

Etiologies of chronic myocardial injury were adjudicated for all 81 patients (**Table 5**). The most common etiologies were CKD (n = 30, 37%), heart failure (n = 23, 28%), and

**Table 5.** Etiologies of chronic myocardial injury in a northern Tanzanian ED, 2020–2023.

| | Chronic Myocardial Injury Patients (N = 81) | |
|---|---|---|
| **Etiology** | **n** | **%** |
| CKD | 30 | 37% |
| Heart Failure | 23 | 28% |
| Hypertension | 7 | 9% |
| Sepsis/SIRS | 4 | 5% |
| Persistent Arrhythmias | 2 | 2% |
| Cardiomyopathy / Structural Heart Disease | 1 | 1% |
| Pulmonary Embolism | 1 | 1% |
| Indeterminate | 13 | 16% |

**Table 6. Causes of death among participants with chronic myocardial injury, northern Tanzania, 2020–2023.**

| | Patients with Chronic Myocardial Injury Dying within Thirty Days (N = 31) | |
|---|---|---|
| **Cause of Death** | **n** | **%** |
| Renal Failure | 10 | 32 |
| Heart Failure | 7 | 23 |
| Pneumonia | 6 | 19 |
| Malignancy | 2 | 6 |
| COPD | 1 | 3 |
| Hyperkalemia | 1 | 3 |
| Liver Cirrhosis | 1 | 3 |
| Opioid Overdose | 1 | 3 |
| Pulmonary Embolism | 1 | 3 |
| Indeterminate | 1 | 3 |

hypertension (n = 7, 9%). The etiology of chronic myocardial injury was indeterminate for 13 (16%) patients.

The causes of death of the 31 participants with chronic myocardial injury who died within thirty days are presented in **Table 6**. The most common causes included renal failure (n = 10, 32%), heart failure (n = 7, 23%), and pneumonia (n = 6, 19%).

## Discussion

This is the first study that we are aware of from SSA to report the characteristics and prognoses of patients with chronic myocardial injury. We found that the presence of stably elevated cTnT values had several cardiovascular and noncardiovascular etiologies and was associated with high rates of mortality among a cohort of patients from northern Tanzania. Our findings are consistent with studies from high-income countries that found chronic myocardial injury to be a heterogeneous condition that predicts adverse prognosis.

The most common comorbidity among patients with chronic myocardial injury was CKD, which was present in 80% of cases. This finding is unsurprising because a majority of patients with end-stage renal disease have elevated cTn values [23, 24], and multiple prior studies from high-income settings have highlighted an association between nonischemic myocardial injury and CKD [25, 26]. Furthermore, over one-third of patients with chronic myocardial injury reported a history of heart failure. This finding is consistent with previous studies from high-income settings that cite an elevated risk for heart failure among patients with stably elevated cTn values [7, 27].

Interestingly, patients with chronic myocardial injury were more likely to be men compared to patients without myocardial injury in our cohort, which may be due to the fact that men tend to have slightly higher serum troponin levels than women [28]. Previous research from high-income countries has showed that the age-specific rates of CVD are higher among men than women in most age groups [29, 30], but the absolute number of women living with CVD is greater due in part to their longer life expectancy [31, 32]. In SSA, research on sex differences in cardiovascular health is limited, but some studies show that women in the region are more likely to have conditions like obesity, hypertension, and ischemic heart disease [33–35]. The higher prevalence of chronic myocardial injury among men in our cohort may be attributed to noncardiovascular factors, but more research is needed on this topic. Another factor associated with chronic myocardial injury in our study was income, with patients with chronic myocardial injury being more likely to earn a lower income than those without

myocardial injury.Socioeconomic disadvantage may have put participants at greater risk for chronic myocardial injury due to complicated access to healthcare, suboptimal dietary and exercise habits, and higher likelihood for behaviors like tobacco and alcohol usage. Ultimately, this finding supports longstanding research on poverty as a key risk factor for CVD globally [36–38]. In SSA in particular, action to address systemic issues like poverty will be needed to curb a rising burden of CVD and mitigate health disparities facing socioeconomically disadvantaged patients throughout the region [39].

Thirty-day mortality rate were significantly higher among patients with chronic myocardial injury than those without any myocardial injury, which is consistent with previous studies from high-income country settings [4, 7, 25, 40]. This is first study to demonstrate chronic myocardial injury as a poor prognostic indicator in SSA, with 38% of patients dying within a short, thirty-day follow-up period. This mortality rate is significantly greater than the 11% reported over six months in a cohort of patients with chronic myocardial injury from the United States [4]; this difference may be due to higher troponin levels in our cohort but may also suggest a substantially higher risk of death for chronic myocardial injury in SSA. Thus, consistent with prior studies from other world regions, our results should alert clinicians to avoid dismissing stably elevated cTn values even without overt myocardial ischemia among patients and to address comorbidities related to worse outcomes [41–43]. Furthermore, thirty-day mortality rates were similar between participants with chronic and acute forms of myocardial injury in our study, which is again consistent with the literature from high-income settings [25, 27, 40]. Although our study did not prospectively follow patients over a long-term period, a study that tracked outcomes over four years among patients with myocardial injury also found no difference in mortality rates between those with chronic and acute injury [27].

This study has several limitations. Firstly, given their complex and sometimes overlapping etiologies, separating between chronic myocardial injury, acute myocardial injury, and Type 1 MI among patients is challenging, and they may even occur simultaneously [44]. Thus, it is possible that our classification of patients lacks sufficient discrimination. This may have biased determined etiologies and calculated mortality rates among patients with chronic myocardial injury. Furthermore, we relied on participant self-report when documenting co-morbidities such as history of hypertension, heart failure, and prior MI, which likely caused an underestimation in the prevalence of each of these conditions in our study cohort. Additionally, all participants in this study were recruited from a single medical center, which limits the generalizability of our results. Finally, the cTn assay used in this study likely has a lower limit of detection that is above the true 99[th] percentile upper reference limit, which likely resulted in some patients with myocardial injury being misclassified as without myocardial injury.

In conclusion, we found that chronic myocardial injury was associated with a heterogenous group of comorbidities and high rates of short-term mortality in a cohort of participants from northern Tanzania. This is the first study to report poor prognosis of chronic myocardial injury in SSA, and should inform clinical care of patients with stably elevated cTn values in the region. Further research is needed to describe chronic myocardial injury burden, risk factors, and outcomes in other settings in SSA, as well to elucidate the reasons for the high mortality rate observed in chronic myocardial injury patients in Tanzania.

## Supporting information

**S1 Table. Echocardiogram results of patients with chronic myocardial injury, northern Tanzania, 2020–2023.**
(DOCX)

**S1 Data. Study data.**
(XLSX)

**S1 File. PLOS inclusivity in global research form.**
(DOCX)

## Author Contributions

**Conceptualization:** Sophie W. Galson, Gerald S. Bloomfield, Julian T Hertz.

**Data curation:** Faraan O. Rahim, Lauren A. Coaxum, Godfrey L. Kweka, Jerome J. Mlangi, Sophie W. Galson, Tumsifu G. Tarimo, Julian T Hertz.

**Formal analysis:** Faraan O. Rahim, Lauren A. Coaxum, Godfrey L. Kweka, Zak Loring, Jerome J. Mlangi, Sophie W. Galson, Gerald S. Bloomfield, Julian T Hertz.

**Funding acquisition:** Julian T Hertz.

**Investigation:** Francis M. Sakita, Godfrey L. Kweka, Zak Loring, Jerome J. Mlangi, Sophie W. Galson, Tumsifu G. Tarimo, Gloria Temu, Julian T Hertz.

**Methodology:** Francis M. Sakita, Godfrey L. Kweka, Jerome J. Mlangi, Sophie W. Galson, Tumsifu G. Tarimo, Gloria Temu, Gerald S. Bloomfield, Julian T Hertz.

**Project administration:** Francis M. Sakita, Godfrey L. Kweka, Jerome J. Mlangi, Tumsifu G. Tarimo, Gloria Temu, Julian T Hertz.

**Resources:** Francis M. Sakita, Godfrey L. Kweka, Jerome J. Mlangi, Sophie W. Galson, Tumsifu G. Tarimo, Gloria Temu, Gerald S. Bloomfield, Julian T Hertz.

**Software:** Julian T Hertz.

**Supervision:** Francis M. Sakita, Gloria Temu, Gerald S. Bloomfield, Julian T Hertz.

**Validation:** Julian T Hertz.

**Visualization:** Julian T Hertz.

**Writing – original draft:** Faraan O. Rahim, Julian T Hertz.

**Writing – review & editing:** Faraan O. Rahim, Francis M. Sakita, Lauren A. Coaxum, Godfrey L. Kweka, Zak Loring, Jerome J. Mlangi, Sophie W. Galson, Tumsifu G. Tarimo, Gloria Temu, Gerald S. Bloomfield, Julian T Hertz.

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
