## [Decision Letter · Decision Letter 0]

14 Nov 2023

PONE-D-23-28179Characteristics and Outcomes of Patients with Symptomatic Chronic Myocardial Injury in a Tanzanian Emergency Department: A Prospective Observational StudyPLOS ONE

Dear Dr. Rahim,

Thank you for submitting your manuscript to PLOS ONE. After careful consideration, we feel that it has merit but does not fully meet PLOS ONE’s publication criteria as it currently stands. Therefore, we invite you to submit a revised version of the manuscript that addresses the points raised during the review process.

We look forward to receiving your revised manuscript.

Kind regards,

Seunghwa Lee

Academic Editor

PLOS ONE

Reviewers' comments:

Reviewer's Responses to Questions

**Comments to the Author**

1. Is the manuscript technically sound, and do the data support the conclusions?

Reviewer #1: Yes

Reviewer #2: Yes

2. Has the statistical analysis been performed appropriately and rigorously? 

Reviewer #1: Yes

Reviewer #2: Yes

3. Have the authors made all data underlying the findings in their manuscript fully available?

Reviewer #1: Yes

Reviewer #2: Yes

4. Is the manuscript presented in an intelligible fashion and written in standard English?

Reviewer #1: Yes

Reviewer #2: Yes

5. Review Comments to the Author

Reviewer #1: 1. Consider rewriting 2nd sentence of introduction to clarify definition of chronic myocardial injury, versus acute myocardial injury.

2. Did the authors have any data available assessing cardiovascular and non-cardiovascular readmissions between chronic myocardial injury and acute myocardial injury?

Reviewer #2: First of all congratulations. I have worked in a sub-saharan hospital and I understand the difficulties in collecting research data in this enviroment.

In your study you evaluated the presence and the evolution of patients who had chronic myocardial injury. This group of patietns are sizable with a prognosis comparable to that of patients with acute myocardial infaction.

I have a few curiosities. First, I would like to know if cardiac ultrasounds are available at the emergency department KCMC hospital. Indeed, patients with chronic myocardial injury are a mixture of cases and cardiac ultrasound would have been extremely useful. If not please add this as a study limitation.

You also reported that patients with chronic myocardial injury have a lower income compared to other cases. Please discuss how a lower income could have influenced your short term results. Finally, your suggestions that clinicians should be carefull with patients with myocardial injury is not true only in your region (line 290) but all over the world.

Please change.

6. PLOS authors have the option to publish the peer review history of their article (what does this mean?). If published, this will include your full peer review and any attached files.

Reviewer #1: No

Reviewer #2: **Yes: **piergiuseppe agostoni

---

## [Author Response · Author response to Decision Letter 0]

28 Nov 2023

November 27, 2023 

Emily Chenette, Ph.D., Editor-in-Chief

PLOS ONE

Dear Dr. Chenette and the PLOS ONE Editorial Board,

Thank you for your review of manuscript PONE-D-23-28179 entitled “Characteristics and Outcomes of Patients with Symptomatic Chronic Myocardial Injury in a Tanzanian Emergency Department: A Prospective Observational Study” submitted for consideration of publication in PLOS ONE. 

We greatly value the knowledgeable comments and suggestions provided by the Reviewers. Following this constructive feedback, we believe that we have significantly improved our manuscript. Responses to all the Reviewers’ comments are included in the following pages.

We have also made minor adjustments to the language of the Methods section of the manuscript to clarify that the cobas h 232 troponin T assay utilized in this study has a lower limit of detection that is above the 99th percentile upper reference limit. 

We have included two versions of the revised manuscript: a clean copy and another version with tracked changes. The latter version shows all the modifications made in red font, allowing you to review the changes point by point. 

We respectfully resubmit our revised manuscript for your consideration. We hope that our paper is now suitable for publication in PLOS ONE. 

Thank you once again for your time and kind consideration of our manuscript.

Sincerely,

Faraan O. Rahim, BS

Duke Global Health Institute 

310 Trent Dr, Durham, NC 27710

Email: faraan.rahim@duke.edu

Reviewers’ Comments

Reviewer #1

Comment 1: Consider rewriting 2nd sentence of introduction to clarify definition of chronic myocardial injury, versus acute myocardial injury.

Response 1: We appreciate this helpful insight. We have now revised the second sentence of the introduction (lines 28-30) to clarify the distinction between chronic myocardial injury and acute myocardial injury: 

“In chronic myocardial injury, cTn levels are persistently but stably elevated, in contrast to acute myocardial injury, where they rise or fall.

Comment 2: Did the authors have any data available assessing cardiovascular and non-cardiovascular readmissions between chronic myocardial injury and acute myocardial injury?

Response 2: We appreciate the insightful question from this reviewer. Unfortunately, data on cardiovascular and non-cardiovascular hospital readmissions were not obtained from the participants in this cohort. The aim of this study was to characterize participants with chronic myocardial injury and compare their thirty-day mortality rates to those of patients without chronic myocardial injury. Therefore, we focused our follow-up on obtaining data on mortality from these research participants as opposed to data on hospital readmission. 

Reviewer #2

Comment 1: First of all congratulations. I have worked in a sub-saharan hospital and I understand the difficulties in collecting research data in this enviroment. In your study you evaluated the presence and the evolution of patients who had chronic myocardial injury. This group of patietns are sizable with a prognosis comparable to that of patients with acute myocardial infaction.

Response 1: We thank the reviewer for their kind words. 

Comment 2: I have a few curiosities. First, I would like to know if cardiac ultrasounds are available at the emergency department KCMC hospital. Indeed, patients with chronic myocardial injury are a mixture of cases and cardiac ultrasound would have been extremely useful. If not please add this as a study limitation.

Response 2: We thank the reviewer for this question. Yes, cardiac ultrasound is available at KCMC, although access remains limited due to cost considerations. In the cohort of 81 participants with chronic myocardial injury, six had an echocardiogram performed. These data are now reported in the new, supplementary S1 Table. In addition, we have revised the manuscript to include mention of S1 Table (lines 217-218): 

“Six (7%) participants with chronic myocardial injury had an echocardiogram performed during their hospital stay; echocardiogram results are summarized in S1 Table.”

Comment 3: You also reported that patients with chronic myocardial injury have a lower income compared to other cases. Please discuss how a lower income could have influenced your short term results. 

Response 3: We appreciate this insight from the reviewer. In response, we have included the following sentence in the results section (lines 284-286) on how lower income may have influenced rates of chronic myocardial injury in our participant cohort:

“Socioeconomic disadvantage may have put participants at greater risk for chronic myocardial injury due to complicated access to healthcare, suboptimal dietary and exercise habits, and higher likelihood for behaviors like tobacco and alcohol usage.”

Comment 4: Finally, your suggestions that clinicians should be carefull with patients with myocardial injury is not true only in your region (line 290) but all over the world. Please change.

Response 4: We thank the reviewer for raising this important point. We agree that clinicians in all world regions should be cautious of chronic myocardial injury. Therefore, we have revised this sentence (lines 298-301) as suggested:

“Thus, consistent with studies from other world regions, our results should alert clinicians to avoid dismissing stably elevated cTn values even without overt myocardial ischemia among patients and to address comorbidities related to worse outcomes.”

---

## [Decision Letter · Decision Letter 1]

14 Dec 2023

Characteristics and Outcomes of Patients with Symptomatic Chronic Myocardial Injury in a Tanzanian Emergency Department: A Prospective Observational Study

PONE-D-23-28179R1

Dear Dr. Rahim,

We’re pleased to inform you that your manuscript has been judged scientifically suitable for publication and will be formally accepted for publication once it meets all outstanding technical requirements.

Kind regards,

Seunghwa Lee

Academic Editor

PLOS ONE

Additional Editor Comments (optional):

Reviewers' comments:

Reviewer's Responses to Questions

**Comments to the Author**

1. If the authors have adequately addressed your comments raised in a previous round of review and you feel that this manuscript is now acceptable for publication, you may indicate that here to bypass the “Comments to the Author” section, enter your conflict of interest statement in the “Confidential to Editor” section, and submit your "Accept" recommendation.

Reviewer #1: All comments have been addressed

Reviewer #2: All comments have been addressed

2. Is the manuscript technically sound, and do the data support the conclusions?

Reviewer #1: Yes

Reviewer #2: Yes

3. Has the statistical analysis been performed appropriately and rigorously? 

Reviewer #1: Yes

Reviewer #2: Yes

4. Have the authors made all data underlying the findings in their manuscript fully available?

Reviewer #1: Yes

Reviewer #2: Yes

5. Is the manuscript presented in an intelligible fashion and written in standard English?

Reviewer #1: Yes

Reviewer #2: Yes

6. Review Comments to the Author

Reviewer #1: Thank you for your resubmission. I believe the reviewer comments have been adequately addressed in this resubmission.

Reviewer #2: No further comments. I have no additional comments for the author or concerns aout the dual publication, research ethics, or publication ethics.

7. PLOS authors have the option to publish the peer review history of their article (what does this mean?). If published, this will include your full peer review and any attached files.

Reviewer #1: No

Reviewer #2: No
